Horizontal transfer generates genetic variation in an asexual pathogen

Huang Xiaoqiu xqhuang@iastate.edu
Department of Computer Science and Plant Sciences Institute, Iowa State University , Ames, IA , USA
Kumar Abhishek
Electronic publication date: 2014 Oct 30
Publication date: 2014
Volume: 2
Electronic Location ID: e650
Received 2014 Jun 26; Accepted 2014 Oct 13
Copyright: © 2014 Huang
Copyright year: 2014
Copyright holder: Huang
License: This is an open access article distributed under the terms of the Creative Commons Attribution License, which permits unrestricted use, distribution, reproduction and adaptation in any medium and for any purpose provided that it is properly attributed. For attribution, the original author(s), title, publication source (PeerJ) and either DOI or URL of the article must be cited.
License URL: https://creativecommons.org/licenses/by/4.0/

Keywords: Asexual reproduction, Horizontal transfer, Nonhomologous recombination, Verticillium dahliae, Lineage-specific regions, Genetic variation

Funding: Iowa State University This work was supported by Iowa State University. The funders had no role in study design, data collection and analysis, decision to publish, or preparation of the manuscript.

==============================
There are major gaps in the understanding of how genetic variation is generated in the asexual pathogen Verticillium dahliae. On the one hand, V. dahliae is a haploid organism that reproduces clonally. On the other hand, single-nucleotide polymorphisms and chromosomal rearrangements were found between V. dahliae strains. Lineage-specific (LS) regions comprising about 5% of the genome are highly variable between V. dahliae strains. Nonetheless, it is unknown whether horizontal gene transfer plays a major role in generating genetic variation in V. dahliae. Here, we analyzed a previously sequenced V. dahliae population of nine strains from various geographical locations and hosts. We found highly homologous elements in LS regions of each strain; LS regions of V. dahliae strain JR2 are much richer in highly homologous elements than the core genome. In addition, we discovered, in LS regions of JR2, several structural forms of nonhomologous recombination, and two or three homologous sequence types of each form, with almost each sequence type present in an LS region of another strain. A large section of one of the forms is known to be horizontally transferred between V. dahliae strains. We unexpectedly found that 350 kilobases of dynamic LS regions were much more conserved than the core genome between V. dahliae and a closely related species (V. albo-atrum), suggesting that these LS regions were horizontally transferred recently. Our results support the view that genetic variation in LS regions is generated by horizontal transfer between strains, and by chromosomal reshuffling reported previously.

Introduction

The prevailing wisdom concerning asexual reproduction is that strictly asexual organisms depend mostly on random mutation to produce genetic variation and lack the genetic innovation required for adaptation to changing environments (Felsenstein, 1974; Burt, 2000; McDonald & Linde, 2002). On the other hand, many important plant pathogens have an apparent long history of asexual reproduction (Masel et al., 1996; Michielse & Rep, 2009). As an example, the vascular wilt fungus Verticillium dahliae is strictly asexual (Usami, Itoh & Amemiya, 2009a; Usami, Itoh & Amemiya, 2009b; Atallah et al., 2010; Inderbitzin et al., 2011). V. dahliae is a soil-borne plant pathogen that can infect hundreds of plant species including economically important crops such as tomato (Pegg & Brady, 2002; Bolek et al., 2005; Fradin & Thomma, 2006).

A study of genetic variation was recently conducted (de Jonge et al., 2013) in a population of ten sequenced V. dahliae strains (de Jonge et al., 2012) together with the genome of V. dahliae strain VdLs17 (Klosterman et al., 2011) as a reference. The study revealed that all strains share a “core genome” of ∼30 Mb, with the rate of single-nucleotide polymorphism (SNP) in the core genome between the reference strain and each of the ten strains ranging from ∼0.2 per kb (strain JR2) to ∼5 per kb (strain St.100). In addition to the core genome, the strains harbor up to 4 Mb of genome sequence that is unique or shared by only a subset of strains, known as lineage-specific (LS) regions, which are enriched for transposable elements (Klosterman et al., 2011; Amyotte et al., 2012). Furthermore, it was revealed by pairwise alignment of the genome assemblies of strains VdLs17 and JR2, and confirmed by PCR that there are chromosomal rearrangements between VdLs17 and JR2, with their syntenic breakpoints correlated with LS regions (de Jonge et al., 2013).

There are major gaps in the understanding of how genetic variation is generated in the asexual pathogen V. dahliae. Besides random mutation and chromosomal reshuffling, it is unknown whether elements with beneficial mutations in LS regions are often transferred horizontally between V. dahliae strains. Here, we analyzed the previously sequenced V. dahliae population of nine strains from various geographical locations and hosts (strain CBS381.66 was excluded; see Methods for details). We found highly homologous elements in LS regions of each strain; LS regions of V. dahliae strain JR2 are much richer in highly homologous elements than the core genome. We also found unexpectedly that 350 kilobases of dynamic LS regions were much more conserved than the core genome between V. dahliae and a closely related species (V. albo-atrum), suggesting that these LS regions were horizontally transferred recently. In addition, we discovered, in LS regions of JR2, several structural forms of nonhomologous recombination, and two or three homologous sequence types of each form, with almost each sequence type present in an LS region of another strain. A large section of one of the forms is known to be horizontally transferred between V. dahliae strains. We determined the mating type of each V. dahliae strain to study the relationship between the mode of reproduction and the extent of variation between strains.

Materials & Methods

Definitions

Recombination between homologous chromosomal segments is called homologous recombination, whereas recombination between nonhomologous ones is called nonhomologous recombination. For nonhomologous recombination, the sequences of two chromosomal segments may differ for most of their lengths, but the fragments at the sites of recombination may be similar. A structural form refers to a sequence structure produced by nonhomologous recombination. The structure consists of two sections combined at the site of recombination.

Short DNA sequences are assembled into sets of overlapping sequences called contigs. A supercontig is a set of contigs ordered and oriented with respect to one strand of the genome. A complete set of supercontigs is called an assembly. After an assembly for a strain is produced, SNPs between this reference strain and other strains can be found by generating much shorter sequences called reads and mapping the reads onto the reference assembly for each of the other strains. Note that there is no biological basis for the notion of reference strain; the reference strain is introduced because of the limitations of the current genome sequencing technology.

Sequence data

All raw read data for ten V. dahliae strains were downloaded from the NCBI Sequence Read Archive (SRA) under BioProject PRJNA169154 (de Jonge et al., 2013). The data for each strain consist of a file of first mates paired with a file of second mates. Strain CBS381.66 was not included in this analysis because its two files of mates were identical. The genome assemblies for two V. dahliae strains and V. albo-atrum strain were downloaded from GenBank. The accession numbers for the 55 supercontigs of the genome assembly of V. dahliae strain VdLs17 are DS572695.1, DS572696.1, DS572697.1, … DS572749.1. Those for the 27 supercontigs of the genome assembly of V. albo-atrum strain VaMs102 are DS985214.1, DS985215.1, DS985216.1, … DS985240.1. Those for the 8 chromosomes of the genome assembly of V. dahliae strain JR2 are CM001863.1, CM001864.1, CM001865.1, …, CM001870.1. Those for the 7 unplaced genomic scaffolds of the JR2 assembly are KE108823.1, KE108824.1, KE108825.1, …, KE108829.1.

SNP detection

SNPs between a set of Illumina paired-end reads and a genome assembly were computed by mapping the reads onto the genome assembly as a reference with Bowtie2 (Langmead & Salzberg, 2012) and calling SNPs with Samtools (Li et al., 2009). The following data and options for Bowtie2 were used: a pair of mate files in fastq format; a minimum insert length of 120 and a maximum insert length of 600; a minimum acceptable alignment score as a function (L, 0, −0.3) of read length in end-to-end mode. The output from Bowtie2 in SAM format was redirected to Samtools with view command and minimum mapping quality of 40 to produce output in BAM format, which was sorted with sort command. The sorted output in BAM format was piled up on the reference with mpileup command and with the following options: −C 50 for downgrading mapping quality for reads containing excessive mismatches; −E for increasing sensitivity; −q 40 for minimum mapping quality; −d 60,000 for maximum number of usable reads at a position. The pileup output was redirected to the Bcftools program with view command and with -vcg option to produce an output file of SNPs in VCF format. The file of SNPs was screened to accept only SNPs with a minimum quality value of 80 and a minimum read coverage depth of 10. An SNP is of type 2 if its max-likelihood estimate of the first ALT allele frequency is between 0.5 (inclusive) and 0.6 (exclusive), and is of type 1 otherwise.

Two custom programs were written for processing files of SNPs in VCF format with the same reference. One program takes as input two files of SNPs and produces as output two files of unique SNPs and a file of common SNPs all in VCF format. The program was repeatedly used to compute a file of SNPs common to several input files of SNPs. The second program takes as input two files of SNPs from strains of the opposite mating types and reports a reference position such that a region immediately upstream of the position is rich in unique SNPs but a region immediately downstream of the position is rich in common SNPs. The program was used to find the site for a potential homologous recombination event in the past.

Detection of recombination forms

The set of Illumina paired-end reads from strain JR2 were mapped with Bowtie2 onto each of the three genome assemblies: the JR2 assembly, the VdLs17 assembly, and the VaMs102 assembly. The three mapped references were correlated by using their syntenic regions. Paired-end reads at breakpoints in the three mapped references were checked for concordant and discordant alignment. Additional sites of nonhomologous recombination were found by processing two files of paired-end reads in SAM format: one produced using the JR2 assembly and the other produced using either the VdLs17 assembly or the VaMs102 assembly. The two files in the order of paired-end reads were screened to produce a subset of the file for the JR2 assembly. A pair of paired-end read from strain JR2 was kept in the subset file if and only if it was concordantly mapped to one genome assembly but discordantly mapped to the other genome assembly. Then the subset file for the JR2 genome assembly in SAM format was sorted by reference coordinates. The sorted subset file was used to find regions of the reference assembly with a significant number (e.g., at least 30) of concordantly mapped pairs of paired-end reads and a significant number of discordantly mapped pairs of paired-end reads. Such a region was the site of recombination for a potential nonhomologous recombination event. Genes in recombination forms were found by performing database searches with Blastx (Gish & States, 1993); transposons in the JR2 and VaMs102 assemblies were identified based on strong sequence similarity (90% sequence identity over a length of 3 kb) to a previously characterized set of transposons in the VdLs17 assembly (Amyotte et al., 2012).

Determination of the mating type of each strain

The amino-acid sequences of the MAT1-1-1 and MAT1-2-1 proteins of V. dahliae were compared to the VdLs17 and VaMs102 genome assemblies with the AAT package (Huang et al., 1997). An amino-acid-DNA alignment of 100% percent identity was constructed between the MAT1-2-1 protein of 156 residues and a region of VdLs17 supercontig 1.4 from positions 356,203 to 356,718 bp with an intron of 48 bp. No match was found between the MAT1-1-1 protein and the VdLs17 assembly. An amino-acid-DNA alignment of 91% percent identity was constructed between the MAT1-1-1 protein of 421 residues and a region of VaMs102 supercontig 1.2 from positions 1,743,583 to 1742,265 bp (in reverse orientation) with an intron of 56 bp. No full-length match was found between the MAT1-2-1 protein and the VaMs102 assembly.

The MAT1-2-1 locus in VdLs17 supercontig 1.4 was checked for coverage by each V. dahliae strain in its read coverage mapping over the VdLs17 reference genome. The locus was completely covered by every V. dahliae strain except DVD-s29. A region of VdLs17 supercontig 1.4 from positions 352,898 to 356,940 bp was not covered by reads from DVD-s29, but its immediately flanking regions of length at least 18,000 bp were completely covered by reads from DVD-s29. Note that the MAT1-1-1 locus in VaMs102 supercontig 1.2 was not covered by reads from DVD-s29 either because VaMs102 and DVD-s29 are different species with an average sequence percent identity of 92%. A de novo assembly for DVD-s29 was performed with an Illumina version of PCAP (Huang et al., 2003). The DVD-s29 assembly contains a contig that has a full-length match with only one residue difference to the MAT1-1-1 protein. The MAT1-1-1 and MAT1-2-1 loci are located in 1.66-Mb regions of VaMs102 supercontig 1.2 and VdLs17 supercontig 1.4, respectively, which were confirmed with DDS2 (Huang et al., 2004) to be syntenic.

Results

SNPs in LS regions

We looked for SNPs in V. dahliae strain JR2. The Illumina reads from a whole genome 500-bp paired-end library of JR2 (de Jonge et al., 2013) were mapped onto the genome assembly of V. dahliae strain VdLs17 (Klosterman et al., 2011) as a reference. The mapping reported a total of 3904 SNPs between JR2 and VdLs17. Of the 3904 SNPs, 873 SNPs revealed that the reads from JR2 at each SNP position in the reference have two alleles: the REF allele and an ALT allele, where the REF allele refers to the allele in the reference and ALT alleles refer to alternate non-reference alleles. These 873 SNPs are of type 2, where an SNP between a strain and the reference is of type 2 if both the REF allele and the ALT allele are present in the strain, and of type 1 if only the ALT allele is present in the strain. To examine the distribution of the 873 type 2 SNPs in the reference, the mapped reference was partitioned into 813 windows each with 35-kb sufficiently covered base positions, and the type 2 SNP rate of each window was calculated. The maximum type 2 SNP rate of all the windows is 2.71 per kb, which is more than sixteen standard deviation units above the mean type 2 SNP rate of 0.03 per kb. The non-random SNP distribution suggests that the SNPs are the result of non-random processes. More importantly, the top eight windows by the SNP rate are all located in LS regions. A manual inspection of the type 2 SNPs in the core genome found that these SNPs almost always occurred in sequence contexts of extended runs of a single base, a known type of context for sequencing errors. For example, an SNP A/C (A, REF allele; C, ALT allele) was reported in the context GTGAAG[A]CCCCCC (with the REF allele enclosed by a pair of square brackets) at position 1,189,199 bp with a coverage depth of 35 in VdLs17 supercontig 1.1.

Of the 873 type 2 SNPs, 612 occurred in LS regions with a combined size of 1.5 Mb, with many of them in sequence contexts without extended runs of a single base. Another manual examination of the 95 SNPs in the top window revealed three sequence types TG (19; T and G were linked by 19 reads), CG (20), and TA (22) in strain JR2, formed by two close type 2 SNPs T/C (T, 58 occurrences; C, 22) and G/A (56/30) at positions 1,033,835 and 1,033,865 bp of VdLs17 supercontig 1.9. After observing the high coverage depths (80 and 86) by reads for the two type 2 SNPs, we found that 64 was the average depth of coverage for the 612 type 2 SNPs in LS regions, 25 was the average depth for the 261 type 2 SNPs in the core genome, and 28 was the average depth for the 3031 type 1 SNPs in the whole genome. In sum, the type 2 SNPs in the core genome were dispersed with low depths of coverage in sequence contexts of extended runs of a single base. In contrast, the type 2 SNPs in the LS regions were concentrated with high depths of coverage often in sequence contexts without extended runs of a single base. This contrast revealed a major difference in the chromosomal structure of JR2 between the core genome and LS regions: the LS regions are rich in highly similar but unidentical sequences, whereas the core is poor in such sequences. In the above example, the three types of highly similar sequences were found in the LS regions of JR2. To see whether the three sequence types were present in the other strains, we examined the read mappings for the other strains at the two positions of VdLs17 supercontig 1.9. Only the sequence type TA (17; T and A were linked by 17 reads) was found in strain St.100, formed by the REF allele T (27) and a type 1 SNP G/A (0/22) at the positions, and only the sequence types TG (23) and CG (22) were found in strain DVD-s26, formed by a type 2 SNP T/C (29/35) and the REF allele G (65) at the positions. Both TG (16) and CG (43) were found in strain St14.01, as in DVD-s26. Only the type CG (15 to 25) was found in strains DVD-3, DVD-31, DVD-161 and DVD-s29. There was no coverage of the region by reads from DVD-s29. Similarly, another example with type 2 SNPs at positions 615,713 and 615,747 bp of VdLs17 supercontig 1.9 illustrates that JR2 was composed of three types of highly similar sequences, with one or two out of the three types found in the other strains. In summary, the type 2 SNPs in LS regions are often in sequence contexts without extended runs of a single base, have a much higher average depth of coverage than the type 2 SNPs in the core genome and the type 1 SNPs in the whole genome, and are mostly due to the presence of at least two types of highly similar sequences in LS regions, not a result of sequencing errors in reads.

We also checked the alleles at the positions of each of the 612 type 2 SNPs in strains St.100 and DVD-s26. St.100 is an outgroup for the rest of the strains in the V. dahliae population, and DVD-s26 is also far distant from JR2 (de Jonge et al., 2013). As is presented below, DVD-s26 is of the same mating type as JR2, and its ancestor branched out from a common ancestor of VdLs17 and JR2 before their current mode of asexual reproduction started. The reads from St.100 and those from DVD-s26 were mapped onto the VdLs17 genome assembly as a reference. Of the 612 SNPs, 529 were selected such that St.100 has the REF allele at each of their positions. Because St.100 is an outgroup, the REF allele at the position of each of these 529 type 2 SNPs is probably the ancestral allele. Of the 529 SNPs, 245 were selected such that DVD-s26 has the ALT allele at each of their positions. In sum, although JR2 is far distant from St.100 and DVD-s26, JR2 shares, with them, many of the REF or ALT alleles at the the positions of the 612 SNPs in the reference.

The next section describes a case where one of the highly similar sequences found with type 2 SNPs in an LS region of JR2 was linked to a previously known horizontal gene transfer event.

Conservation of LS regions within and between species

We examined the conservation of an LS region within V. dahliae strains, which are divided into two groups: race 1 and race 2. Race 1 strains have an effector named AVE1 that activates resistance in tomato with the VE gene; race 2 strains lack the effector (de Jonge et al., 2012). In the V. dahliae population, only JR2 and St14.01 are race 1 strains (strain CBS381.66 is not counted). The AVE1 gene is located in a 51.4-kb section of an LS region of JR2 chromosome 5 (CM001867). The 51.4-kb section is highly conserved between JR2 and St14.01 (de Jonge et al., 2012). This was confirmed by mapping the reads from St14.01 onto the JR2 genome assembly produced by de Jonge et al. (2013). A total of 72,599 SNPs were found between the St14.01 reads and the JR2 assembly, corresponding to a genome-wide SNP rate of ∼2.17 per kb. The 51.4-kb section was almost completely covered by St14.01 reads. To see if a reference base in the JR2 assembly is confirmed by reads from JR2, we mapped the reads from JR2 onto the JR2 assembly. There was only one position in the 51.4-kb section such that the JR2 and S14.01 reads differ at the position; the JR2 reads support the reference base, whereas the St14.01 reads support an alternate base. Thus, only one SNP (C/A at position 690,180 bp of JR2 chromosome 5) is present between St14.01 and JR2 in the 51.4-kb section. As is described below, the 51.4-kb section is part of one of the four recombination forms in JR2. The conservation of the 51.4-kb section between St14.01 and JR2 suggests an exchange of the effector AVE1 between the V. dahliae strains, after horizontal transfer (de Jonge et al., 2012) of AVE1 to a V. dahliae strain from a plant.

The 51.4-kb section is connected at its 3′ end to a 5.6-kb section beginning at position 690,580 bp of JR2 chromosome 5, with both sections completely inside an LS region. The 5.6-kb section was almost completely covered in high depth by reads from JR2: its average depth of coverage was 83, more than twice as large as the average depth (41) of the 51.4-kb section. In the read coverage of the 5.6-kb section, 28 type 2 SNPs were found, revealing at least three types of highly similar sequences. In addition, the 5.6-kb section is also highly conserved between JR2 and St14.01, as indicated by 24 common type 2 SNPs in the 5.6-kb section and two common types of sequences between the two strains; the average depth of coverage by reads from St14.01 was 39 over the 51.4-kb section and 51 over the 5.6-kb section. For example, three sequence types AGC (linked by 20 reads), ACT(17) and GGC (21) were found in the JR2 read coverage of a 33-bp fragment in the 5.6-kb section, formed by three close type 2 SNPs A/G (55/28), G/C (66/20) and C/T (61/21) at positions 691,591, 691,617 and 691,623 bp of JR2 chromosome 5. Only two types AGC (17) and ACT(21) were found in the St14.01 read coverage of the fragment, formed by REF allele A, type 2 SNP G/C (23/21) and type 2 SNP C/T (21/21) at the three positions of JR2 chromosome 5. An important observation about multiple types of highly similar sequences in JR2 LS regions, as revealed by type 2 SNPs in JR2 read coverages, is that one of such sequences was linked to a horizontal gene transfer into the JR2 lineage, through the physical connection between the 51.4-kb section and the 5.6-kb section.

Although LS regions are unique to a subset of V. dahliae strains, some of them are more conserved than the core genome between V. dahliae and V. albo-atrum. A database search using Blastx (Gish & States, 1993) with the DNA sequence (as a query) of an LS region from strain JR2 surprisingly found that the JR2 sequence was more similar to a protein sequence from V. albo-atrum strain VaMs102 than any protein sequence from V. dahliae strain VdLs17. Because of this search result, we also mapped the JR2 reads onto the VaMs102 assembly (Klosterman et al., 2011). We found that the core genome of the VaMs102 assembly was hardly covered by JR2 reads, but some LS regions (with a combined length of 350 kb) of supercontigs 1.14, 1.19 and 1.20 in the assembly were densely covered in high depth by JR2 reads with a nucleotide sequence identity mostly above 99%. This was totally unexpected as the core genome is more much conserved among Verticillium species than LS regions; V. albo-atrum strain VaMs102 and V. dahliae strain VdLs17 share an average 92% nucleotide sequence identity over the core genome (Klosterman et al., 2011). This finding suggests that the LS regions were exchanged between species V. albo-atrum and V. dahliae. To see which of the V. dahliae strains is most similar to V. albo-atrum strain VaMs102 over the LS regions, we mapped the reads from each of the other V. dahliae strains onto the VaMs102 assembly. Strains JR2 and St.100 each have at least 24% more read coverage of the VaMs102 LS regions than the other V. dahliae strains; St.100 has 6% more read coverage than JR2. The results suggest that the LS regions in strain VaMs102 came from an ancestor that was more closely related to St.100 than to the other strains.

Structural variations in LS regions

We also sought chromosomal-structure variations in strain JR2. Four structural forms of LS elements were found in the JR2 Illumina reads that were mapped to a region (referred to as form 1) of JR2 chromosome 5, to a region (form 2) of VaMs102 supercontig 1.19, to a region (form 3) of VaMs102 supercontig 1.20, or to an initial section of form 2 combined with a terminal section of form 1 (form 4). Forms 1 through 4 are shown in Fig. 1. Note that some of these JR2 reads were mapped to a 11.3-kb region of VdLs17 supercontig 1.9, whose reverse complement is almost identical to an initial half of form 2. Only form 1 is present in the JR2 genome assembly; form 2 and parts of forms 3 and 4 have no large-scale similarity to the JR2 assembly.

Figure 1 Four structural forms of LS elements produced by nonhomologous recombination.

Each form is shown as a combination of two sections (denoted by color lines) at the site of recombination (denoted by a black rectangle); each gene/domain is denoted by a white rectangle and each transposon by a gray rectangle. Except for form 4, above each form is the reference chromosome or supercontig in which the form is located. On the right are the specific combinations of forms in each strain, as indicated by lines of the same color. Each form is supported by concordantly mapped 500-bp JR2 read pairs (denoted by a pair of angle arrows above the black rectangle) linking the two sections across the site of recombination. Except for form 4, the 5′ position of each recombination site in the reference chromosome or supercontig is given, as pointed by an upward arrow below the black rectangle. Form 4 shares the VdLINE1section with form 2, and the BRCT1 section with form 1. Gene/domain/transposon names: AVE1, avirulence on VE1 tomato; BRCT, the carboxy-terminal domain of a breast cancer susceptibility protein in cell cycle checkpoint functions responsive to DNA damage; ANK, ankyrin repeat and protein kinase domain-containing protein; CYC, protein binding domain functioning in cell-cycle and transcription control; VdLTRE1, LTR retrotransposon of superfamily Copia in the V. dahliae genome; VdLINE1, non-LTR retrotransposon of superfamily 1 in the V. dahliae genome.

A comparison of the four structural forms revealed that they resemble results produced by nonhomologous recombination. Forms 1, 2 and 4 share nearly identical 320-bp segments (with at most one nucleotide substitution) at their sites of recombination; form 3 has a 280-bp segment at the site of recombination that is similar to the 3′ portion of the 320-bp segment with 11 nucleotide substitutions and with no gaps. Note that the length of the common segment in each form is at least 180 bp smaller than the average insert size of 500 bp for paired-end reads. In addition, each of forms 1 through 3 has a unique section, and form 4 has a unique combination of sections. Thus, although a read completely inside the common segment can have matches to more than one form, a pair of paired-end reads (read pair) outside and across the common segment can not have matches to more than one form. On the basis of these observations, the read pairs with both reads mapped onto the sections of a form outside and across its common segment were used as evidence to support the presence of the form in the reads. In particular, form 4 was found to be present in the reads of JR2 solely based on read pairs; form 4 was not found in any of the three assemblies as a reference. For each form, the number of read pairs that support the form, along with the 5′ position of the common segment in the reference sequence, is given in Fig. 1. One notable observation is that form 3 was supported by much more read pairs than the other forms. The explanation is that the common segment in form 3 is 280 bp long, 40 bp shorter than that in the other forms, causing more 500-bp read pairs to cross it.

To see if any of the forms could be found in the other strains, we mapped reads from each of the other strains onto each of the three assemblies as a reference. Each of forms 1 to 3 was found in some of the other strains (Fig. 1); form 4 was found only in JR2. Some notable observations are given as follows. First, both forms 2 and 3 were found in St.100 and VaMs102. Since VaMs102 is a different species and St.100 is an outgroup to the rest in the V. dahliae population, forms 2 and 3 probably occurred before form 4, which shares the RT section with form 2. Then, form 1 with the AVE1 section was found only in JR2 and St14.01, with only one SNP between JR2 and St14.01 in the AVE1 section, as indicated previously. Forms 1 and 4 share the BRCT1 section as well as the site of recombination. Thus, form 1 was probably produced recently by nonhomologous recombination from form 4. Next, form 1 was linked to a horizontal transfer event involving the AVE1 gene. Forms 2 and 3 were parts of LS regions horizontally transferred between V. dahliae and V. albo-atrum, as indicated in the previous section. Finally, forms 1, 3 and 4 all contain a BRCT domain involved in cell cycle checkpoint functions responsive to DNA damage (Bork et al., 1997), where the BRCT1 section in form 1 and the BRCT2 section in form 3 share a nucleotide sequence identity of 98.7%. Our analysis linked the BRCT domains to nonhomologous recombination in Verticillium, whereas they are known to maintain genome stability in homologous recombination.

Forms 1, 2 and 4 all contain a retrotransposon; form 3 is 6 kb to one. Although transposable elements are prevalent in the JR2 and VdLs.17 genomes, few transposons were found in the VaMs.102 genome (Klosterman et al., 2011; Amyotte et al., 2012; de Jonge et al., 2013). The retrotransposons associated with forms 2 and 3 in the VaMs.102 genome are two of the only four transposons identified in the genome.

We examined the conservation of form 3 between the VaMs102 genome assembly and the reads from strain St.100. A large part of form 3 (a 7,320-bp region from positions 38,571 to 45,890 bp of VaMs102 supercontig 1.20) was mostly covered by concordantly mapped reads from St.100. The region was searched for SNPs and was found to contain only one SNP. Thus, the region is remarkably conserved between the two species, considering a genome-wide SNP rate of 2.99 per kb between strains St.100 and VdLs17 of the same species.

SNPs were used to determine how different sequence types of form 3 were distributed in strains JR2, St100, St14.01 and DVD-s26. Note that form 3 is located in VaMs102 supercontig 1.20. SNPs were sought in the coverages of form 3 by reads from each of the four V. dahliae strains. Four base positions in form 3 were found, at each of which two alleles occurred in the JR2 read coverage. For example, consider position 40,089 bp of VaMs102 supercontig 1.20, one of the four positions in form 3. The REF allele A at this position was confirmed by 43 concordantly mapped reads from strain St.100. However, a type 2 SNP A/C (58/34) was found in the JR2 read coverage of this position with 89 concordantly and 3 discordantly mapped reads. Interestingly, only the ALT allele (C) was observed in the DVD-s26 read coverage of this position with 24 concordantly and 7 discordantly mapped reads, and in the St14.01 read coverage of this position with 19 concordantly and 5 discordantly mapped reads. The same kind of observation was made at each of the three other positions of form 3 (Table 1). In particular, we found that for each read that spans the last two positions (26 bp apart), the read has either both REF alleles or both ALT alleles at the positions; no REF allele at one position is linked by a read to the ALT allele at the other position. These observations indicate that both sequence types (REF and ALT) of form 3 are present in JR2. Note that the REF sequence type of form 3 is the same as form 3, which is from VaMs102 supercontig 1.20. Only the REF sequence type was found in St.100; only the ALT sequence type was found in St14.01 and DVD-s26. Because St14.01 and DVD-s26 are more closely related than JR2 is to either of them, and St.100 is an outgroup to the three strains (de Jonge et al., 2013), the REF sequence type in St.100 and VaMs102 was probably the ancestral type.

Table 1 Alleles in four strains at the positions of four SNPs in form 3.a

SNP	SNP	REF/ALT	REF/ALT allele count in strain	
Position (bp)	Context	Allele	JR2	St.100	St14.01	DVD-s26	
40,089	TTATT[A]GCGAC	A/C	58/34	43/0	0/24	0/31	
40,875	GGCGG[C]GTTGC	C/T	37/36	28/0	0/31	0/33	
41,161	GAAGG[G]TATCG	G/C	18/52	31/0	0/34	0/21	
41,187	GCACA[A]AGGTG	A/G	23/52	25/0	0/24	0/17	
Notes.

a Form 3 is part of supercontig 1.20 of V. albo-atrum reference strain VaMs102.

Similar observations were made with 12 SNPs in form 2 (Table 2). Although form 2 is not completely present in the reads of V. dahliae strain St14.01 or strain DVD-s26, part of the VdLINE1 section of form 2 was covered by the reads of both St14.01 and DVD-s26. The observations indicate that two sequence types (REF and ALT) of the VdLINE1 section are present in JR2. Only the REF sequence type, probably the ancestral sequence type, is present in St.100; only the ALT sequence type is in St14.01 and DVD-s26. Each REF allele is present in a large number of reads in JR2 & St.100, and so is each ALT allele in JR2, St14.01 & DVD-s26. Thus, it is unlikely that these alleles are the result of sequencing errors. These observations show that the strains differ in sequence type composition.

Table 2 Alleles in four strains at the positions of 12 SNPs in form 2.a

SNP	SNP	REF/ALT	REF/ALT allele count in strain	
Position (bp)	Context	Allele	JR2	St.100	St14.01	DVD-s26	
174,943	CGCGG[A]CACAA	A/C	50/31	18/0	0/57	0/51	
175,798	CCACC[A]CACCA	A/G	38/36	40/0	0/62	0/57	
175,849	CCAAG[G]CCGAC	G/A	49/32	41/0	0/59	0/48	
175,901	ATCTC[T]ACGGC	T/C	36/37	43/0	0/67	0/62	
176,672	AGGCC[C]ATCTC	C/A	34/25	33/0	0/29	1/52	
176,710	CGAGT[G]GGTTG	G/C	31/30	31/0	0/35	0/48	
176,906	GTGTG[C]AAAGA	C/T	37/38	34/0	0/78	0/76	
176,953	AGGAG[G]GTTGC	G/A	39/42	49/0	0/67	0/83	
177,636	TCATC[G]CGATA	G/A	48/39	28/0	0/39	0/74	
177,676	CATCG[T]CAGTG	T/C	53/40	31/0	0/58	0/76	
178,443	AGACG[A]CTACC	A/G	23/15	34/0	0/55	0/56	
178,521	TACAA[C]AACCT	C/T	16/17	15/0	0/40	0/47	
Notes.

a Form 2 is part of supercontig 1.19 of V. albo-atrum reference strain VaMs102.

None of the four forms is present in the reads of V. dahliae strain DVD-s29; the site of recombination and nearby sections of each form were sparsely covered by discordantly mapped reads from DVD-s29. The same observation was made for each of strains DVD-3, DVD-31, DVD-161 and DVD-s94, which form a separate cluster (de Jonge et al., 2013).

Conservation of SNPs in a separate cluster of strains

We explored genetic conservation and variation in the separate cluster of strains DVD-3, DVD-31, DVD-161 and DVD-s94. When the reads from each strain were mapped onto the VdLs17 reference assembly of 33.9 Mb, it was found that the four strains share 53,653 SNPs, differing from VdLs17 at the rate of 1.58 common SNPs per kb. On the other hand, the strains in the cluster are much closer to each other; the rate of unique SNP between each pair of strains is between 0.4 and 0.7 per kb. Very few of these SNPs are of type 2; the numbers of type 2 SNPs in each strain are 172 (DVD-3), 82 (DVD-31), 160 (DVD-161), and 199 (DVD-s94). The four strains in the cluster share 60 type 2 SNPs (Table S1). Of these, 50 were concentrated in the LS region of VdLs17 supercontig 9.1, and the rest were in the LS region of VdLs17 supercontig 8.1. Of the 50 type 2 SNPs, 30 occurred in protein-coding regions, of which 13 were synonymous and 17 were nonsynonymous. The concentration of the 50 common type 2 SNPs in the LS region of supercontig 9.1 indicates that each strain is composed of multiple types of highly similar sequences that differ in LS regions and that the variations in sequence type are conserved in the cluster.

Then we checked if any of the sequence type variations is present outside the cluster. None of the 50 type 2 SNPs was found in strain St.100. Only one of the 50 type 2 SNPs (at position 615,747 bp of VdLs17 supercontig 9.1) is shared by strains St14.01, JR2, DVD-s26 and DVD-s29, indicating that the corresponding sequence type variation is conserved in these strains outside the cluster. The SNP occurred in the coding region of a short new hypothetical protein (of 53 residues) with 83% percent identity to hypothetical protein VDBG_10182; it was a synonymous substitution at the third base of the codon for residue 52. Another of the 50 SNPs (at position 1,041,375 bp) is shared by strains JR2 and DVD-s26. A third of the 50 SNPs (at position 614,488 bp) was found in strain JR2. None of the 50 type 2 SNPs was found as type 1 in any strain except DVD-s29, indicating that the sequence types in strains St14.01, St.100, JR2 and DVD-s26 contain no ALT alleles at the positions of all or most of the 50 SNPs, so the REF allele at each position was the ancestral allele for the cluster. On the other hand, 7 of the 50 type 2 SNPs were found as type 1 in strain DVD-s29; the sequence types in DVD-s29 contain only the ALT alleles at the positions of these 7 SNPs. Note that for each of the 50 type 2 SNPs, each strain in the cluster has one sequence type with the REF allele and another sequence type with the ALT allele at the SNP’s position. For each of the 7 SNPs, strain DVD-s29, unlike any strain in the cluster, contains only the sequence type with the ALT allele at the position of the SNP. This observation, combined with the information (given below) that all V. dahliae strains except DVD-s29 have the same mating type, supports that many of the REF and ALT alleles were present during the period of sexual reproduction and that the strains in the cluster or their ancestors acquired their REF or ALT alleles at the positions of these SNPs by exchanging DNA over LS regions with other strains of the same mating type.

Next we estimated negative selection pressure on the LS region with the 50 SNPs by considering the level of sequence conservation in the LS region between V. dahliae strain VdLs17 and V. albo-atrum strain VaMs102. When the VaMs107 assembly instead of the VdLs17 assembly was used as the reference, 43 type 2 SNPs in the LS region of VaMs102 supercontig1.19 were shared by all the four strains in the cluster. Of the 43 type 2 SNPs, 42 are on the list of the 50 type 2 SNPs. This was unexpected, considering that the average nucleotide identity of the VdLs17 and VaMs102 genomes is about 92% (Klosterman et al., 2011). The 42 SNPs occurred in a section of VaMs102 supercontig1.19 from positions 174,514 to 212,281 bp. The VaMs102 section is syntenic to a section of VdLs17 supercontig1.9 from positions 1,169,722 to 1,208,190 bp. The level of conservation between the two sections was assessed by using V. dahliae strain St.100, an outgroup to all the other V. dahliae strains (de Jonge et al., 2013). The overall SNP rate between St.100 and VdLs17 is 2.99 per kb. The SNP rate between St.100 and VdLs17 over the section of VdLs17 supercontig 1.9 is 0.75 per kb. The SNP rate between V. dahliae strain St.100 and V. albo-atrum strain VaMs102 over the section of VaMs102 supercontig 1.19 is 0.71 per kb. In other words, over the LS section where the 42 SNPs occurred, St.100 is equally close to both VdLs17 and VaMs102. However, over the whole genome, strain St.100 is much closer to VdLs17 than VaMs102, as is seen by mapping the reads from St.100 to both VdLs17 and VaMs102 assemblies. The number of paired-end reads from St.100 was 6,136,331. Of these, 76.79% were aligned concordantly exactly once to the VdLs17 assembly, and only 13.88% were aligned concordantly exactly once to the VaMs102 assembly. Thus, the LS section is highly conserved after the exchange of DNA segments containing the LS section between V. dahliae and V. albo-atrum. Note that the LS section is part of a 97-kb LS segment conserved between VdLs17 and VaMs102.

The extent of variation between strains reveals their mode of reproduction

We determined the mating type of V. albo-atrum strain VaMs102 and each V. dahliae strain by using the sequences of the V. dahliae MAT1-1 and MAT1-2 genes (Usami, Itoh & Amemiya, 2009a; Usami, Itoh & Amemiya, 2009b). Of the ten V. dahliae strains including VdLs17, nine have the MAT1-2 gene and only DVD-s29 has the MAT1-1 gene. Strain VaMs102 has the MAT1-1 gene (Klosterman et al., 2011). The MAT1-1 and MAT1-2 genes share the same site in the genome, so only one of them can be present in the genome, indicating that V. dahliae and V. albo-atrum were heterothallic in the past. For example, the MAT1-2 locus is at positions 348 to 358 kb of supercontig 1.4 of V. dahliae strain VdLs17; the MAT1-1 locus is at positions 1737 to 1747 kb of supercontig1.2 of V. albo-atrum strain VaMs102. The regions surrounding the two loci are syntenic.

The nine strains are almost identical over the coding region in the MAT1-2 locus for the MAT1-2-1 protein of 156 residues. Strains VdLs17 and St.100 have only a synonymous substitution at the third base of the codon for residue 72; VdLs17 and JR2 have only a nonsynonymous substitution at the first base of the codon for residue 75. The other six strains are identical to St.100 over the coding region.

SNPs between reference strain VdLs17 and other V. dahliae strains were used to investigate the relationship between reproduction mode and genetic variation. This was made possible by the finding that strain DVD-s26 has the MAT1-2 gene and strain DVD-s29 has the MAT1-1 gene. Both DVD-s26 and DVD-s29 were collected from soil in Canada (Essex County) in 1994 (de Jonge et al., 2012). The group of SNPs between DVD-s26 and VdLs17 is referred to as the DVD-s26 group, and the DVD-s29 group is defined similarly. The two groups have 55,525 common SNPs; the DVD-s26 group has 15,999 unique SNPs and the DVD-s29 group has 17,167 unique SNPs. Thus, there are 33,166 (17,167 + 15,999) SNPs (at the rate of ∼1 SNP per kb) between DVD-s26 and DVD-s29 with the opposite mating types. Those SNP numbers show that DVD-s26 and DVD-s29 are more closely related than VdLs17 is to either of them (de Jonge et al., 2013). Although strains VdLs17 and JR2 were collected from different hosts in different geographical locations: VdLs17 from lettuce in USA (California) and JR2 from tomato in Canada (de Jonge et al., 2012), the number (3904) of SNPs between JR2 and VdLs17 is much lower than that (33,166) between DVD-s26 and DVD-s29 from soil in the same county. A possible explanation for these observations is that no sexual reproduction took place in strains JR2 and VdLs17 since their separation; sexual reproduction continued in strains DVD-s26 and DVD-s29 after their separation from VdLs17, which produced many of the differences between DVD-s26 and DVD-s29.

SNPs were used to find a potential homologous recombination event in the past. A large variation in the number of SNPs unique to DVD-s26 or DVD-s29 was found in two adjacent regions with their common border at position 400,000 bp of VdLs17 supercontig1.3. The region of length 70 kb immediately upstream of the border contains 52 SNPs unique to DVD-s26, 7 SNPs unique to DVD-s29, and 2 SNPs common to both strains. On the other hand, the region of length 100 kb immediately downstream of the border contains 3 SNPs unique to DVD-s26, 4 SNPs unique to DVD-s29, and 14 common SNPs. The variation in the numbers of common and unique SNPs indicates a potential recombination event in the core genome between DVD-s26 and DVD-s29.

Discussion

The prevailing wisdom concerning asexual reproduction is that strictly asexual organisms lack mechanisms to produce genetic variation and cannot adapt to changing environments. However, this view cannot explain why V. dahliae switched from the past mode of sexual reproduction to the present mode of asexual reproduction to become adaptable to hundreds of plant species. Here, we investigated the genetic variation in a V. dahliae population with both mating types from different geographical locations. Our results show that asexual reproduction is divided into two main modes of reproduction: clonal and aclonal. Aclonal reproduction generates genetic variation through horizontal recombination, a nonsexual mechanism involving horizontal transfer and chromosomal reshuffling of which a special case is nonhomologous recombination. Evidence for genetic variation in the form of multiple types of homologous sequences was found in the sequence reads of every V. dahliae strain. The finding of the low SNP rate between some of the strains in the core genome confirms that they propagate by asexual reproduction at the present time (de Jonge et al., 2013). Because LS regions are enriched for repetitive sequence elements (Klosterman et al., 2011; Amyotte et al., 2012), horizontal recombination is likely mediated by transposition.

Aclonal reproduction complements sexual reproduction by generating homologous sequences with single-nucleotide and structural differences in LS regions. Sexual reproduction produces, by random mutation and homologous recombination, genome-wide variation. The past sexual reproduction in V. dahliae had produced significant genome-wide diversity among strains. On the other hand, aclonal reproduction generates a significant depth of localized variation by horizontal recombination. The current aclonal reproduction in V. dahliae has produced both point and structural changes in LS regions in the same strain. The finding of a conserved LS region in V. albo-atrum strain VaMs102 and the finding of common types of sequences in DVD-s26, St14.01 and JR2 suggest that LS regions are transferred between strains and between species. The diversity among strains in the core genome and homologous sequences in the LS regions of the same strain might be what V. dahliae needs to become adaptable to hundreds of plant species.

The finding of multiple types of homologous sequences in the LS regions of the same strain provides support for balancing selection. Each of three of the four recombination forms in strain JR2 was found in another strain or species, so the form was viable on its own. In addition, multiple sequence types of a recombination form were found in the same strain. Multiple types of homologous sequences were conserved in a cluster of four strains. Possible explanations for maintaining multiple types of homologous sequences in the same strain are that they have a higher adaptive value than any single type of sequence and that this benefit is achieved without increasing the genome size significantly. Note that multiple types of homologous sequences are limited to smaller LS regions; the larger core genome lacks such sequences. The lack or shortness of a diploid phase in this asexual fungus and the constraint on its genome size might have caused the fungus to come up with the current strategy for achieving some benefits of a diploid organism.

Our study provides evidence at the nucleotide level for mitotic genetic exchange known as parasexual cycle. Evidence for the parasexual cycle in laboratory populations of V. dahliae was reported (Pulhalla & Mayfield, 1974; O‘Garro & Clarkson, 1992). The hypothesis that parasexuality facilitates the exchange of effector genes between asexual lineages was proposed (Noguchi, Yasuda & Fujita, 2006; Chuma et al., 2011). Although parasexuality was thought to be virtually excluded in nature by a genetic mechanism (known as vegetative incompatibility) that restricts heterokaryon formation between asexual lineages (Glass, Jacobson & Shiu, 2000), an isolate was found by using DNA markers to carry two alleles at each of several loci, viewed as evidence for mitotic recombination, in a field population of the asexual rice pathogen Magnaporthe oryzae (Zeigler et al., 1997). Parasexuality is a mode of aclonal reproduction. Our results show that horizontal recombination in field populations of V. dahliae generates genetic variation mostly in LS regions; lack of type 2 SNPs in the core genome indicates that no horizontal recombination took place in the core genome. In addition, our results confirm that horizontal recombination is commonly used in nature to produce genetic variation. As LS regions are known to harbor effector genes (de Jonge et al., 2013), our study suggests that horizontal recombination facilitates the exchange of effector genes in LS regions between asexual lineages.

Our results raise the possibility that recombination between DNA molecules is a nearly universal mechanism for generating genetic variation in reproduction. There are two complementary types of recombination: recombination between homologous chromosomes in sexual reproduction and recombination between nonhomologous segments in aclonal reproduction. Homologous recombination occurs within the same species and generates slightly different recombinant chromosomes, making it suitable for gradual evolution in stable environments. Nonhomologous recombination, on the other hand, can occur between different species, and generates hugely different recombinant segments, making it suitable for rapid adaptation to changing environments.

The analysis of the genetic variation in the V. dahliae strains shows that every strain has its unique combination of homologous sequences in LS regions. We speculate that the unique combination of homologous sequences in LS regions provides the genetic variation needed for the adaptation of the strain to its environment. Genetic variation is generated in V. dahliae by horizontal transfer and incorporation of small LS elements into chromosomes. In contrast, genetic variation in Fusarium oxysporum is generated by horizontal transfer of whole LS chromosomes (Ma et al., 2010). For this reason, the genome of Fusarium oxysporum can be twice as large as that of V. dahliae. It remains to be seen whether the mechanism for generating genetic variation in V. dahliae is used in other filamentous pathogens such as Phytophthora infestans (Haas et al., 2009).

The difference in the rate of type 2 SNP between the core genome and LS regions can be used to check on the neutral theory of molecular evolution (Kimura, 1983) which claims that most of the changes in the genetic material are caused by random drift of mutant alleles that are neutral or nearly neutral. Genetic drift is the same random evolutionary force operating both on the core genome and on LS regions. However, the above difference cannot be explained by the view that genetic drift is the chief cause of molecular evolution. Thus, the difference supports the view that mutation and natural selection are the chief cause of molecular evolution. The difference also reveals that the core genome and LS regions are subject to different types of mutational and selectional processes. The mutational process in the core genome is known as random mutation; the mutational process in LS regions is called horizontal recombination. Nonhomologous recombination is involved in exchange of LS regions between strains. The conservation of multiple types of homologous sequences in different strains serves as evidence for balancing selection in LS regions.

Conclusions

We have performed an analysis of genome sequence data from a V. dahliae population and V. albo-atrum strain VaMs102, with both species being strictly asexual plant pathogens. Our results revealed that LS regions are highly enriched for homologous sequences, whereas the core genome lacks this feature. In particular, the LS regions of V. dahliae strain JR2 are comprised of homologous sequences with single-nucleotide differences and with various structural forms like those produced by nonhomologous recombination. Some of the sequence types and structural forms in JR2 were found in other V. dahliae strains or the V. albo-atrum strain. The extraordinary conservation of some homologous sequences in LS regions between V. dahliae strains and between the two species, as compared to that in the core genome, suggests that those homologous sequences were acquired by horizontal transfer, consistent with a previous finding of a horizontal transfer event (de Jonge et al., 2012). Building on the previous discovery of chromosomal rearrangements in V. dahliae (de Jonge et al., 2013), we conclude that genetic variation in V. dahliae LS regions is generated by horizontal transfer and chromosomal reshuffling between strains and between species. Understanding the evolutionary mechanism of LS regions is an important step in finding an effective way of controlling V. dahliae.

Supplemental Information

Table S1 60 Common type 2 SNPs in strains DVD-3, DVD-31, DVD-161 and DVD-s94

Click here for additional data file.

The author thanks the Plant Sciences Institute and the Laurence H. Baker Center for Bioinformatics & Biological Statistics for providing Linux clusters on which this work was conducted. The author is grateful to all the people who have made suggestions and comments on this manuscript.

Additional Information and Declarations

Competing Interests

Author Contributions

The author declares there is no competing interests.

Xiaoqiu Huang conceived and designed the experiments, performed the experiments, analyzed the data, contributed reagents/materials/analysis tools, wrote the paper, prepared figures and/or tables, reviewed drafts of the paper.

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
