# Peer review of "Horizontal transfer generates genetic variation in an asexual pathogen"

_PeerJ, doi:10.7717/peerj.650_

## Round 0.1 · original submission · Minor Revisions

This MS is a highly detailed SNP analysis which provides important inside in the evolution of Verticillium dahliae. Please address the comments of the reviewer #1, before we formally accept your MS.

·

Basic reporting

This article adheres to PeerJ standards.
It is written in clear English.
The title sounds a little overstated since the paper describes only ONE pathogen while the title may suggest a broader conclusion. I would strongly advise a more precise title.
The article should for sure include some kind of a legend/explanation since the author uses so many definitions: forms, groups, reference strain/assembly etc.. Sometimes it is hard to get through; what is what. :)
The structure is OK.
In my opinion a better figure should present LSs, forms etc., putting it in a broader context. For example, what are the specific combinations of forms in each strain? How do they look? Are they similar? Does the author find repetitive elements and are those transposons (which is not stated in the paper)? I suggest some kind of a map showing similarities, differences and transfers of fragments.

Experimental design

This work clearly defines the problem.
The method is a very clever idea.
The work seems to be conducted rigorously and nicely, but the method and definitions are described partly in Methods section and partly at the beginning of the Results section. Maybe some basic facts may be moved or at least repeated into Methods?

Validity of the findings

The data is sound and seems to be controlled throughout the process.
As for the Discussion section, I have a few questions/comments that may help the author to improve the article:
- The author states in lines 475-6: "to become adaptable to hundreds of plant species." However he does not explain why so. Why asexual reproduction favours such a broad pathogenecity?
- lines 485-7: The author suggests an influence of transposons, but there are no data that shows the presence of transposons in LS regions;
- at the end of Discussion section the author says that nonhomologous recombination is responsible for LS exchange. On the other hand his discoveries show that these regions are homologous. So why isn't it a homologous recombination while we suspect the involvement of HGT...?
- I see no broader view on the subject. Why the author, with the new tools he built, did not try his method on other pathogens to have a broader perspective? This would strongly enhance his findings.

Additional comments

I read your article with interest and I find it to touch an important point. This type of work needs a broader analysis now to see ahether it is specific or general for eukaryotic pathogens. Congratulations.

I would however suggest a few minor changes that I did not include in the previous sections of the review since there was no space for it:
- Introduction, line 40: you suggest HGT as the only source of variation. Why? Single mutations may also play a role;
- line 97: were mapped not was mapped;
- line 108: "sufficient number" - what do you mean?
- line 151: "were generated": is this an appropriate term in here? Later you suggest an evolutionary term. ;)
- line 159: you find that 30% of SNPs are outside LS regions. I think this finding may lead to a very interesting analysis: either these 30% are dispersed or concentrated in specific areas of the genome. If the latter is true, this may be a crucial finding of novel LS regions or any other specific regions;
- lines 187-8: "Taken together, we conclude that most type 2 SNPs in LS regions are a result of highly similar sequences in LS regions, not a result of sequencing errors in reads". How SNPs may be a result of highly similar sequences? This sentence should be more precise;
- lines 357-8: "These observations show that the variation among strains in sequence type composition contributes to genetic variation in the strains". It sounds like a tautology...
- lines 383-4: a hypothetical protein is suggested but no name/id is given;
- lines 385-90: It would be interesting to learn whether SNPs are more likely to locate in coding or non-coding regions. Is such a c/nc rate important?
- lines 434-437: "For example, the MAT1-2 locus is at positions 348 to 358 kb of supercontig 1.4 of V. dahliae strain VdLs17; the MAT1-1 locus is at positions 1737 to 1747 kb of supercontig1.2 of V. albo-atrum strain VaMs102". Is this info useful in here?
- line 527: "supports" what?
- throughout the article you use the form "we" while there is only one author. I would suggest changing it into "I";
- More generally, I find the notion of reference strain important, however a constant reference to it may dim a bit a broader view on the subject where all strains in fact are equal.

Reviewer 2 ·

Basic reporting

The manuscript is written in clear and good English language. There is sufficient background and introduction, to understand the basics of the manuscript. There is only one figure in the manuscript. All other data are related to the analysis of SNPs.

Experimental design

The manuscript is solely based on SNP analysis. However, as far as I do understand the method it seems to be sound.

Validity of the findings

I do not understand the SNP methods well enough to be fully sure whether all data are valid. However, the author addresses related problems in the manuscript and it appears that these issues are addressed.

Additional comments

This is a highly detailed SNP analysis which provides important inside in the evolution of Verticillium dahliae. The data indicate that genetic variation in
LS regions appears to be generated by horizontal transfer between strains, and also via chromosomal reshuffling. I particularly like the discussion.

---

## Round 0.2 · accepted · Accept

Congratulations, this MS is accepted now.

·

Basic reporting

.

Experimental design

.

Validity of the findings

.

Additional comments

The author accepted and remodeled all major aspects, except for the we/I aspect. So I do accept changes made by the author.

---

## Author Rebuttal · Round 0.2

**Dear Editor and Reviewers,**

**I really appreciate your help with reviewing this manuscript and am grateful to you all for your suggestions that have significantly improved this manuscript. Please see my responses in boldface below.**

Thank you for your submission to PeerJ. I am writing to inform you that in my opinion as the Academic Editor for your article, your manuscript "Acquisition of homologous elements contributes to genetic variation in an asexual pathogen" (#2014:06:2312:0:0:REVIEW) requires some minor revisions before we could accept it for publication. The comments supplied by the reviewers on this revision are pasted below. My comments are as follows:

Editor's comments This MS is a highly detailed SNP analysis which provides important inside in the evolution of Verticillium dahliae. Please address the comments of the reviewer #1, before we formally accept your MS. If you are willing to undertake these changes, please submit your revised manuscript (with any rebuttal information*) to the journal within 45 days.

* A rebuttal letter and any tracked changes can be added to the file uploads page under the "Revision response files" section. Please also upload a clean untracked version for production purposes to the primary files section and replace your previous manuscript. Accepted formats for the rebuttal letter and tracked changes are: docx (preferred), doc, and PDF. As you previously uploaded a single manuscript file for your initial submission you will need to upload any primary high resolution image and table files separately if you have not already done so.

Abhishek Kumar
Academic Editor for PeerJ

**I would like to thank you for your decision and comments. Please note that I had been granted an extension for submitting a revised version beyond the deadline.**

Reviewer Comments
Reviewer 1 (Marcin Grynberg)

Basic reporting

This article adheres to PeerJ standards. It is written in clear English. The title sounds a little overstated since the paper describes only ONE pathogen

while the title may suggest a broader conclusion. I would strongly advise a more precise title.

**I appreciate this advice and have selected a more precise title.**

The article should for sure include some kind of a legend/explanation since the author uses so many definitions: forms, groups, reference strain/assembly etc.. Sometimes it is hard to get through; what is what. :)

**A number of definitions have been added at the start of the Materials & Methods section (lines 56-71 in the revised version).**

The structure is OK. In my opinion a better figure should present LSs, forms etc., putting it in a broader context. For example, what are the specific combinations of forms in each strain? How do they look? Are they similar? Does the author find repetitive elements and are those transposons (which is not stated in the paper)? I suggest some kind of a map showing similarities, differences and transfers of fragments.

**I am grateful for the suggestion. A new figure has been made to give an overview of LSs in reference supercontigs and a close view of each form, showing genes/transposons in each form, common sections in forms, and specific combinations of forms in each strain.**

Experimental design
This work clearly defines the problem. The method is a very clever idea. The work seems to be conducted rigorously and nicely, but the method and definitions are described partly in Methods section and partly at the beginning of the Results section. Maybe some basic facts may be moved or at least repeated into Methods?

**The descriptions of the method have been moved to the Methods section (lines 113-117).**

Validity of the findings The data is sound and seems to be controlled throughout the process. As for the Discussion section, I have a few questions/comments that may help the author to improve the article:

**Your questions/comments are very helpful for me to improve the manuscript.**

- The author states in lines 475-6: "to become adaptable to hundreds of plant species." However he does not explain why so. Why asexual reproduction favours such a broad pathogenecity?

**On lines 513-524, it is suggested that genome-wide variation produced by past sexual reproduction and localized variation produced by horizontal recombination in present asexual reproduction may contribute to such a broad pathogenecity.**

- lines 485-7: The author suggests an influence of transposons, but there are no data that shows the presence of transposons in LS regions;

**The methods used for analysis of transposons/genes have been added (lines 128-132). The results reported on lines 340-344 show that each of the four forms is closely linked to a transposon.**

- at the end of Discussion section the author says that nonhomologous recombination is responsible for LS exchange. On the other hand his discoveries show that these regions are homologous. So why isn't it a homologous recombination while we suspect the involvement of HGT...?

**This is clarified by adding the definitions of homologous and nonhomologous recombination on lines 57-63.**

- I see no broader view on the subject. Why the author, with the new tools he built, did not try his method on other pathogens to have a broader perspective? This would strongly enhance his findings.

**This is a great suggestion. I think that horizontal transfer plays a significant role in generating genetic variation in asexual reproduction. However, I was not able to find more publicly available data from many isolates of asexual pathogens. When the data become available, I will examine them.**

Comments for the author I read your article with interest and I find it to touch an important point. This type of work needs a broader analysis now to see whether it is specific or general for eukaryotic pathogens. Congratulations.

**I really appreciate that you took time reading this paper and thought the topic is important. Your encouragement will allow me and other scientists to see how authors and reviewers can work together on this difficult scientific journey.**

I would however suggest a few minor changes that I did not include in the previous sections of the review since there was no space for it: - Introduction, line 40: you suggest HGT as the only source of variation. Why? Single mutations may also play a role;

**I agree that HGT is not the only source of variation and have made this clear by beginning the sentence with the phrase "Besides random mutation and chromosomal reshuffling" (line 39).**

- line 97: were mapped not was mapped;
   **The correction has been made.**

- line 108: "sufficient number" - what do you mean?
   **The word "sufficient" has been replaced by "significant" to indicate that both the number of concordantly mapped paired-end reads and the number of discordantly mapped paired-end reads are so large that it is unlikely that those paired-end reads occur in the same region by chance.**

- line 151: "were generated": is this an appropriate term in here? Later you suggest an evolutionary term. ;)
   **I agree that the term is not appropriate and have revised the sentence as "... the SNPs are the result of non-random processes." (line 173)**

- line 159: you find that 30I think this finding may lead to a very interesting analysis: either these 30If the latter is true, this may be a crucial finding of novel LS regions or any other specific regions;
   **I have checked the distribution of the type 2 SNPs outside the LS regions and found that they are dispersed, except 14 SNPs in a region of supercontig 1.21 from positions 470,324 to 474,914. The majority of the 14 SNPs have a depth of coverage from 20 to 36.**

- lines 187-8: "Taken together, we conclude that most type 2 SNPs in LS regions are a result of highly similar sequences in LS regions, not a result of sequencing errors in reads". How SNPs may be a result of highly similar sequences? This sentence should be more precise;
   **The texts before this sentence in this paragraph provide evidence for the conclusion. I have replaced the last sentence by the following one: "In summary, the type 2 SNPs in LS regions are often in sequence contexts without extended runs of a single base, have a much higher average depth of coverage than the type 2 SNPs in the core genome and the type 1 SNPs in the whole genome, and are mostly due to the presence of at least two types of highly similar sequences in LS regions, not a result of sequencing errors in reads." (lines 209-213)**

- lines 357-8: "These observations show that the variation among strains in sequence type composition contributes to genetic variation in the strains". It sounds like a tautology...

**The sentence has been reworded as "These observations show that the strains differ in sequence type composition." (line 382)**

- lines 383-4: a hypothetical protein is suggested but no name/id is given;

**It has been indicated that the protein is 83% identical to hypothetical protein VDBG_10182. (lines 407-409)**

- lines 385-90: It would be interesting to learn whether SNPs are more likely to locate in coding or non-coding regions. Is such a c/nc rate important?

**I think so. In the previous paragraph, it was indicated that 30 out of the 50 type 2 SNPs were found in coding regions (a c/nc rate of 30/20). (line 398)**

- lines 434-437: "For example, the MAT1-2 locus is at positions 348 to 358 kb of supercontig 1.4 of V. dahliae strain VdLs17; the MAT1-1 locus is at positions 1737 to 1747 kb of supercontig1.2 of V. albo-atrum strain VaMs102". Is this info useful in here?

**The information might be useful for other scientists to confirm our work and to study the evolution of the MAT1-1/MAT1-2 locus in many V. dahliae strains.**

- line 527: "supports" what?

**I have toned this down by replacing "supports" by "suggests" and adding "in LS regions" (lines 552-553).**

- throughout the article you use the form "we" while there is only one author. I would suggest changing it into "I";

**In Acknowledgements, I have replaced "we" by "the author". I have also checked the two positions on this issue and found that single authors in scientific publications often prefer "we" over "I".**

- More generally, I find the notion of reference strain important, however a constant reference to it may dim a bit a broader view on the subject where all strains in fact are equal.

**I agree. The notion of reference strain was introduced because of the limitation of the genome sequencing/assembly techniques. If we could produce a good assembly for every strain, then there would be no need to use the notion of reference strain.**

Reviewer 2

Basic reporting

The manuscript is written in clear and good English language. There is sufficient background and introduction, to understand the basics of the manuscript. There is only one figure in the manuscript. All other data are related to the analysis of SNPs.

Experimental design

The manuscript is solely based on SNP analysis. However, as far as I do understand the method it seems to be sound.

Validity of the findings

I do not understand the SNP methods well enough to be fully sure whether all data are valid. However, the author addresses related problems in the manuscript and it appears that these issues are addressed.

Comments for the author

This is a highly detailed SNP analysis which provides important inside in the evolution of Verticillium dahliae. The data indicate that genetic variation in LS regions appears to be generated by horizontal transfer between strains, and also via chromosomal reshuffling. I particularly like the discussion.

**I am grateful for your support and comments.**